# A 12-Week, Randomized, Double-Blind, Placebo-Controlled Study to Evaluate the Efficacy and Safety of *Lactobacillus plantarum* LMT1-48 on Body Fat Loss

**DOI:** 10.3390/nu17071191

**Published:** 2025-03-28

**Authors:** Sung-Bum Lee, Byungwook Yoo, Chaemin Baeg, Jiae Yun, Dong-wook Ryu, Gyungcheon Kim, Seongok Kim, Hakdong Shin, Ju Hee Lee

**Affiliations:** 1Department of Family Medicine, Soonchunhyang University Bucheon Hospital, Bucheon 22972, Republic of Korea; sblee@schmc.ac.kr; 2Department of Family Medicine, Soonchunhyang University Seoul Hospital, Seoul 04401, Republic of Korea; dryoo@schmc.ac.kr; 3Global Medical Research Center, Seoul 06526, Republic of Korea; 4Gwanggyo R&D Center, Medytox Inc., Suwon 16506, Republic of Korea; jiae.yun@medytox.com (J.Y.); dwryu@medytox.com (D.-w.R.); 5Department of Food Science & Biotechnology, College of Life Science, Sejong University, Seoul 05006, Republic of Korea; paulkc12@gmail.com (G.K.); skim01@sejong.ac.kr (S.K.); hshin@sejong.ac.kr (H.S.); 6Carbohydrate Bioproduct Research Center, College of Life Science, Sejong University, Seoul 05006, Republic of Korea; 7Department of Dermatology & Cutaneous Biology Research Institute, Yonsei University College of Medicine, Seoul 03722, Republic of Korea

**Keywords:** *Lactobacillus plantarum*, probiotics, obesity, body fat loss

## Abstract

Objectives: This study aims to evaluate the efficacy and safety of probiotics for body fat reduction in obese individuals. Methods: A total of 106 participants with a body mass index between 25 and 30 kg/m^2^ were randomly assigned to either the experimental group treating with *Lactobacillus plantarum* LMT1-48 or the placebo group in the placebo-controlled clinical trial. Body composition was assessed by dual-energy X-ray absorptiometry and computed tomography. Fecal samples between the groups were contrasted via DNA sequencing for evaluation of the microbiota and its diversity. Results: After 12 weeks of follow-up period, the body fat mass decreased significantly, from 30.0 ± 4.4 to 28.3 ± 4.1 kg in the experimental group (*p* = 0.009). The percentage of body fat in the two groups showed a similar trend (*p* = 0.004). Conclusions: LMT1-48 also positively influenced the microbial taxa linked to obesity analyzed by gut microbiome sequencing. LMT1-48 is a safe and collaborative agent to reduce obesity.

## 1. Introduction

Obesity is defined as a condition in which excess body fat accumulates in adipose tissue as a result of excess energy intake, with the remainder of the energy used for metabolic activity being converted to triglycerides [1]. It is influenced by various factors, broadly categorized into environmental factors—such as poor dietary patterns and excessive calorie and carbohydrate intake—and genetic factors, which are being studied for potential obesity-related genes. For example, the fat mass and obesity-associated gene (FTO) and the leptin gene (LEP) influence metabolic diseases and hormonal regulation, contributing to obesity [2,3].

The prevalence of obesity among adults more than 19 years old in South Korea has continued to increase since exceeding 38.4% in 2021, with 49.2% in men and 27.8% in women, which is a significant increase from 30.2% in 2012 [4]. In addition, the socioeconomic costs of obesity have more than doubled in the last decade, and the rate of increase in the socioeconomic costs of obesity (2.22 times) is faster than those of smoking (1.62 times) and alcohol consumption (1.56 times) [5]. In the future, the socioeconomic burden of obesity is estimated to be even greater when the costs of formal obesity management such as diet foods and exercise are included as well as the intangible costs such as reduced quality of life and distress [6]. According to the World Health Organization (WHO), obesity has been identified as the leading cause of human mortality, even surpassing smoking as the leading cause of death [7]. Several recent studies have shown that obese people are at a very high risk of developing chronic diseases such as diabetes, atherosclerosis, liver disease, and cancer [8,9].

A weight loss of 5–10% can reduce cardiovascular risk factors and the risk of developing diabetes; therefore, reducing body weight by 10% is recommended as a common criterion for obesity treatment. If conservative treatments such as diet and exercise do not result in a loss of 10% of the existing body weight after 3–6 months, medication may be considered [10]. However, although pharmacologic treatment of obesity has been tried for a long time, it often comes with several side effects such as headache, dizziness, nausea, and vomiting [11,12]. Therefore, it is necessary to develop materials with less of a risk of side effects.

The *Lactobacillaceae* family makes up the largest group of microorganisms included in probiotics. Within this family, *Lactobacillus plantarum* is particularly adaptable to the environment, being found in soil, plants, food, and feces, unlike other lactobacilli. *L. plantarum* is not only listed on the list of ingredients that can be used in food by the Korea Food and Drug Administration (KFDA), but is also used as an ingredient for lactic acid bacteria in Korean health functional foods [13].

*L. plantarum* LMT1-48 (LMT1-48), the investigational product (IP) for our trial, is a strain isolated from home-made kimchi. In vitro studies have shown that LMT1-48 is involved in adipocyte differentiation and lipogenesis-related enzymes, which are involved in body fat accumulation [14]. Based on these basic studies, in vivo studies using a mouse model of obesity induced by a high-fat diet have shown that LMT1-48 reduced the expression of adipocyte differentiation-related genes and reduced the area of abdominal visceral and subcutaneous fat by micro-CT [14]. This suggests that weight loss and visceral fat reduction through inhibition of body fat synthesis have anti-obesity effects.

Therefore, our double-blind randomized controlled trial study was designed to evaluate the efficacy and safety of LMT1-48 on body fat reduction in obese individuals compared with a control diet. Furthermore, we performed DNA sequencing, microbiota analysis, and diversity assessments on participants’ fecal samples to investigate the enrichment of specific microbial taxa linked to obesity modulation.

## 2. Materials and Methods

### 2.1. Study Design

This was a double-blind randomized controlled trial (RCT). The trial was conducted as a parallel study with randomization to the experimental group who received LMT1-48 or the placebo group, and the required number of subjects was 120 subjects, 60 subjects in each group, to account for the dropout rate (25%). Using block randomization with allocation codes, the experimental group and the placebo group were matched 1:1. All participants meeting the inclusion/exclusion criteria were assigned to each group by the allocation codes. In order to guarantee balanced randomized assignment, each group had an equal number of participants. The randomization table was made by applying permutations of random numbers (A, B random numbers) generated by the randomization program of the SAS^®^ system sequentially, starting from the number 1, of the participants and was pre-designed and generated before launching the clinical trial through SAS^®^. In order to maintain double blinding, the assignment of the unique codes (i.e., information on double blinding) was kept under seal. No unblinding events occurred during the trial. The participants were provided with the IP that matched their randomization number, and we ensured that spares were available in case of defects or breakage of the IP so that blinding could be maintained. Participants in the experimental group were instructed to take two capsules of 500 mg (total amount of LMT1-48 is 5 × 10^9^ CFU) once a day for 12 weeks while drinking water [15,16]. Participants in the placebo group consumed visually the same placebo capsules (500 mg of maltodextrin per capsule). Over a 12-week period, individuals were instructed to return every three weeks to report any side effects, return any unused capsules, and obtain a refill in order to track compliance.

### 2.2. Study Population

Subjects were eligible when they were at least 19 years of age and less than 70 years of age with a body mass index (BMI) of at least 25 kg/m^2^ and less than 30 kg/m^2^ at both Visit 1 and Visit 2 (Appendix A). Individuals with any of the following criteria were excluded: (1) Currently being treated for a serious cardiovascular, immunologic, gastrointestinal, hepatic and biliary, respiratory, urinary, renal, neurologic, infectious, or psychiatric disease. (2) Diagnosed with cancer within 5 years of Visit 1. (3) Used medications that affect weight within 1 month (30 days) of Visit 1. (4) Received antibiotics or antiseizure medications within 2 weeks of Visit 1. (5) Administered probiotics, prebiotics, or a probiotic product on a continuous basis (4 or more times per week) within 2 weeks of Visit 1. (6) Bariatric surgery within 1 year at Visit 1. (7) Patients with uncontrolled hypertension (systolic blood pressure greater than 160 mm Hg or diastolic blood pressure greater than 100 mm Hg). (8) Patients with diabetes who have a fasting blood glucose of 126 mg/dL or greater or are taking antidiabetic medications (e.g., oral glucose-lowering agents, insulin). (9) TSH ≤ 0.1 μIU/mL or ≥10 μIU/mL. (10) Creatinine greater than or equal to 2 times the upper limit of normal. (11) Alanine aminotransferase (AST) or aspartate aminotransferase (ALT) greater than or equal to three times the upper limit of normal for the site. (12) Alcohol intake-induced disorders or central nervous system disorders. (13) Anyone with a musculoskeletal disorder that prevents them from exercising. (14) Weight change of 10% or more within 3 months of Visit 1. (15) Participated in a commercial program for obesity within 3 months of Visit 1. (16) Participation in another clinical trial within 3 months of Visit 1 or plans to participate in another clinical trial after the start of our study. (17) Pregnant or nursing women or those who plan to become pregnant during this human clinical trial. (18) Individuals with severe gastrointestinal disorders who have trouble consuming IP. (19) Individuals who are sensitive or allergic to any of the ingredients in IP. (20) Anyone deemed unsuitable by the principal investigator (PI) for any other reason. Random assignments were used to place the 120 participants in the LMT1-48 and placebo groups. After 7 participants in the experimental group and 7 in the placebo group left the study due to unsuccessful follow-up efficacy evaluation, a total of 106 participants finished the RCT: 53 in the LMT1-48 group and 53 in the placebo group were still left (Appendix A). At Visits 2 and 3, all participants were educated to reduce food intake by 500 kcal per day and to increase daily physical activity by 300 kcal during the study.

### 2.3. Measures of Efficacy

In order to determine the primary efficacy assessment, dual energy X-ray absorptiometry (DEXA) was used to compare the changes in body fat mass before and after medication use. DEXA passes two energy levels of X-rays through the body and measures the difference in radiation absorption by tissues to determine body fat mass, body fat percentage, and lean mass. An accurate body composition analysis that can distinguish and analyze fat and lean mass in different parts of the body was provided by the equipment (Horizon Wi, Hologic Inc., Marlborough, MA, USA). Changes in body weight, thigh circumference, arm circumference, hip circumference, waist circumference, waist/hip ratio, BMI, lean body mass and body fat percentage by DEXA, visceral fat, subcutaneous fat, total abdominal fat area, and visceral/subcutaneous fat area ratio, measured by computed tomography (TOSHIBA SCANNER Activation 1, TOSHIBA, Tokyo, Japan), were among the secondary efficacy evaluations. Blood chemistry profiles (i.e., total cholesterol, HDL cholesterol, LDL cholesterol, triglyceride, AST, ALT, BUN, Creatinine, and WBC), free fatty acid, adiponectin, and fecal flora were also included as the secondary efficacy. Blood samples were collected from the antecubital vein. Blood chemistry profiles were obtained by enzymatic methods by using an automated chemistry analyzer (Cobas C702, Roche, Bavaria, Germany), and free fatty acid was measured by a Hitachi analyzer (Hitachi 7600, Hitachi, Tokyo, Japan). Adiponectin was obtained using a dedicated ELISA kit (Phoenix Pharmaceuticals, Burlingame, CA, USA). Every measurement was performed while fasting. All documented adverse events that started or got worse after the participants were put on IP were examined in order to evaluate safety. Blood pressure and heart rate were measured by the sphygmomanometer (BPBIO 320T, InBody Co., Ltd., Seoul, Republic of Korea). Height and body weight were measured by the equipment (BSM330, InBody Co., Ltd., Seoul, Republic of Korea). BMI was computed as weight divided by height squared. The following information was collected: surgery history within 3 months, comorbidities identified by physicians, and demographic information (age, birth date, and sex). Medication history was checked for all prior medications within 1 month (30 days) as of Visit 1. Concomitant medications and concomitant therapies were identified at Visit 3 and Visit 4 following the consumption of the IP. The Global Physical Activity Questionnaire (GPAQ) was used to gather lifestyle data, including physical activity. GPAQ assesses activity levels in four areas: work, transportation, recreation, and sedentary behavior. This questionnaire captures the intensity, frequency, and duration of physical activities, enabling an accurate analysis of the participant’s physical activity level. Smoking history, divided into non-smoker, ex-smoker, and current smoker, and alcohol intake (more or less than once a month) were collected through self-questionnaires. To calculate the calorie intake, the participants were instructed on the way to fill out dietary survey questionnaires in order to gather information about their eating patterns (frequency of overeating per week, regular meal: yes/no, mealtime: <10/10–20/≥20 min). Dietary intake was assessed by asking the participants to record their food intake for three days (including at least one weekend day) during the previous week. The recorded data were then analyzed using the Can-Pro program of the Korean Nutrition Society to calculate the average daily calorie intake. The PI examined the answered form used to measure the subjects’ daily energy consumption during Visits 2, 3, and 4. At each visit, the individual received information about diet and physical activity.

### 2.4. Collection of Fecal Samples and 16S rRNA Gene-Based Sequencing for Gut Microbiota Analysis

Fecal samples were collected from all the participants during Visits 2 and 4 using a fecal sampling kit (CJ Bioscience, CLSB-02, Seoul, Republic of Korea) and stored in a deep freezer until analysis. Sample preparations and sequencing steps for fecal DNA were performed according to the Earth Microbiome Project (EMP) [17]. Total genomic DNA was extracted from fecal samples using the Power Soil-htp 96-Well Soil DNA Isolation Kit (Qiagen Inc., Redwood City, CA, USA) according to the manufacturer’s instructions. The V4 region of 16S rRNA genes was amplified with 515F/926R primers [18], and the amplicons were subjected to sequencing using the Illumina MiSeq platform (2 × 300 cycles, paired-end; San Diego, CA, USA).

### 2.5. Evaluation of Safety Set

Subjects who were randomized to the clinical trial and consumed the IP at least once were chosen as the safety set. The safety set included 60 participants in the LMT1-48 group and 60 participants in the placebo group. Furthermore, complete blood count, biochemical parameters, vital signs, and adverse events were analyzed for evaluating safety.

### 2.6. Statistical Analysis

Intra-group comparisons of changes in body fat mass and body fat percentage before and after IP intake by DEXA were analyzed using paired *t*-tests. The degree of change between the experimental and placebo group was assessed for statistical significance by Wilcoxon rank sum tests or two-sample *t*-tests, depending on whether normality was met. If the association of groups in demographics with groups in lifestyle surveys were statistically significant, ANCOVA was performed with the relevant baseline characteristics as covariates. The other efficacy endpoints were body weight; BMI; thigh, arm, waist, and hip circumference; waist/hip circumference ratio; lean body mass by DEXA; subcutaneous fat area, visceral fat area, and total abdominal fat area by CT; blood biochemistry; and free fatty acid and adiponectin. Intra-group comparisons of before and after the intake were analyzed by paired *t*-tests, and the degree of change between the experimental and placebo groups was evaluated by two-sample *t*-tests or Wilcoxon rank sum tests, depending on whether normality was met. In order to examine the confounders for categorical variables, chi-square or Fisher’s exact tests were conducted. If relationships between groups in demographic and lifestyle surveys were statistically significant, ANCOVA was conducted with the relevant baseline characteristics as covariates.

For analysis of the fecal microbiota, the sequencing data were processed using Quantitative Insights into Microbial Ecology (QIIME2) (2023.02 version), and the analyses were performed at the Biopolymer Research Center for Advanced Materials (BRCAM, Sejong University, Seoul, Republic of Korea) [19]. The amplicon sequence variant (ASV) tables were generated by demultiplexing sequences and performing quality control using the DADA2 method [20]. All ASV sequences were aligned with MAFFT v7 [21] and used to construct a phylogeny with FastTree2 [22]. The taxonomy of each ASV was determined using the SILVA database (v119). Alpha diversity, calculated by Faith’s Phylogenetic Diversity; observed feature counts and Shannon entropy index; and beta diversity, calculated by unweighted UniFrac distance metrics, were measured using q^2^-diversity [23]. For statistical analysis, the non-parametric Kruskal–Wallis test and the permutational multivariate analysis of variance (PERMANOVA) were used to determine significant differences in microbial diversity and bacterial structures, respectively [24,25]. Linear discriminant analysis effect size (LEfSe) analysis (LDA score) was used to identify the most significant differences in bacterial taxa [26].

Statistical significance was defined as a *p*-value of less than 0.05. SAS^®^ was used to conduct statistical analyses (Version 9.4, SAS Institute, Cary, NC, USA).

## 3. Results

### 3.1. Clinical Study Results

Baseline characteristics of the participants are tabulated in Table 1. For the experimental group, 53 middle-aged Koreans with an average age of 44.6 ± 12.9 with height of 166.3 ± 8.3, weight of 75.5 ± 8.2, and BMI of 27.3 ± 1.3 kg/m^2^ participated, and for the placebo group, 53 participants with an average age of 40.7 ± 10.2, height of 166.7 ± 9.3, weight of 75.3 ± 9.6, and BMI of 27.0 ± 1.3 kg/m^2^ participated (Table 1). Moreover, there was no statistically significant difference in the intake of IP between the case and control groups (*p* = 0.590), with the case group’s mean compliance being 96.18 ± 7.47% and the control group’s being 95.36 ± 8.04%. The two groups did not vary statistically significantly in any of the baseline measures except systolic blood pressure (SBP), diastolic blood pressure (DBP), and visceral fat area.

The changes in anthropometric measurements such as weight, BMI, arm circumference, waist circumference, hip circumference, calorie intake, and physical activity are displayed in Table 2. The experimental group revealed weight loss of 0.90 ± 1.33 kg after 6 weeks (*p* < 0.001) and 1.41 ± 1.79 kg after 12 weeks (*p* < 0.001), which is higher than the placebo group, showing weight loss of 0.56 ± 1.70 kg after 6 weeks (*p* = 0.021) and 1.13 ± 2.06 kg after 12 weeks (*p* < 0.001). But it was insignificant compared between the groups. BMI, arm circumference, waist circumference, and hip circumference found similar trends. The participants in both groups showed significant reduced calorie intake during the trials as educated, but no significant difference was found between the groups. The physical activity was not significantly increased in both groups.

In Table 3, the changes in radiologic measurements and biochemical profiles are tabulated. Compared with the placebo group, the experimental group suggested a decrease in body fat mass and body fat percentage from baseline to after 12 weeks, which is statistically significant. Body fat mass decreased significantly in the experimental group by 1.6 ± 1.9 kg compared with the placebo group, where it decreased by 0.7 ± 2.2 kg (*p* = 0.009). Moreover, body fat percentage decreased significantly in the case group by 1.5 ± 2.0% compared with the placebo group, where it decreased by 0.4 ± 2.1% (*p* = 0.004). On the other hand, lean body mass increased by 0.2 ± 1.4 kg after 12 weeks in the experimental group, while lean body mass decreased by 0.4 ± 13 kg in the placebo group, which is statistically significant between groups. Moreover, higher reductions in visceral fat area, subcutaneous fat area, and total abdominal fat area were shown in the experimental group than the placebo group, although there was no statistical significance between the two groups. There were no significant differences in the other secondary endpoints acquired by blood biochemical analysis (Table 3).

Body fat mass and body fat percentage were measured for each body part by DEXA (Table 4). The body parts were divided into head, trunk, android, gynoid, both arms, and both legs. Comparing the results from baseline to 12 weeks, the experimental group was significantly reduced in body fat mass of the trunk, android, gynoid, and both legs. In other words, body fat was significant lost in all areas except the head and both arms. Additionally, the change in body fat percentage was significantly decreased in the experimental group compared with the placebo group in all body parts except the head.

### 3.2. Fecal Microbiota Analysis

A total of 4,862,644 sequence reads was obtained from the fecal samples, with an average of 22,937 ± 7453 sequence reads per sample. These reads were binned into ASVs (Appendix A).

To evaluate the impact of probiotic treatment on gut microbial diversity, alpha diversity was assessed using Faith’s Phylogenetic Diversity, observed feature counts, and Shannon’s index. Across all alpha diversity metrics, no significant changes in microbial richness or evenness were observed between pre- and post-treatment in both experimental and placebo groups (Figure 1A–C). Additionally, no significant alterations in bacterial communities were observed between pre- and post-treatment in either group, as evaluated by weighted and unweighted UniFrac distance (Figure 1D,E).

However, post-treatment intra-group distances based on unweighted UniFrac significantly increased in both the placebo and experimental groups, suggesting greater variability in microbiota composition among subjects after treatment (Figure 1F). For weighted UniFrac distances, post-treatment intra-group distances significantly increased in the experimental group but decreased in the placebo group. This indicates that the microbiota structure became more variable among the subjects in response to probiotic treatment in the experimental group, while it remained more consistent in the placebo group (Figure 1F).

Consistent with stable gut microbial composition and structure in both experimental and placebo groups (Figure 1), no changes were observed in the relative abundance of the top 20 bacterial taxa (Figure 2A). The Firmicutes to Bacteriodetes (F/B) ratio was not changed significantly in both groups or between groups (Appendix A). To further dissect the bacterial taxa classified as minor taxa, we analyzed the relative bacterial abundance at the genus level using LEfSe. In the placebo group, two genera—*Clostridium_T* and *Lactococcus*_A_346120—were underrepresented post-treatment. In contrast, the experimental group showed a significant increase in *Morganella* and *Lactobacillus*, while *Clostridium_T* and *Eubacterium*_O_258270 were underrepresented pre-treatment (Figure 2B; LDA > 2.0). Comparing post-treatment differences between the experimental and placebo groups, five bacterial genera, *Hafnia*, *Anaerococcus*, *Lactobacillus*, *Lachnospira*, *and Phocaeicola*_A_858004, were significantly enriched in the experimental group. Meanwhile, several taxa—including *Phascolarctobacterium_A*, *Parasutterella*, *Eubacterium* G, *Gastranaerophilaceae*_g_CAG196, *Lachnospiraceae* UBA5905, *Eubacterium*_O_258270, *Aldercreutzia*_404257, and *Lachnospira*_g_CAG_95, were underrepresented in the experimental group compared with the placebo group (Figure 2B).

### 3.3. Safety Assessment

Adverse event rates throughout the 12-week trial period were not significantly associated between the two groups; the LMT1-48 group had an adverse event rate of 8.3% (5/60 participants), while the placebo group had an adverse event rate of 11.7% (7/60 participants) (Appendix A). No significant adverse effects were noted. There were no serious adverse events and no dropouts due to adverse events. In addition, there were no significant differences in the safety sets acquired by blood analyses (Appendix A).

## 4. Discussion

When compared with the placebo group, 12-week consumption with LMT1-48 significantly decreased body fat mass and body fat percentage and enhanced lean mass through our trial. Furthermore, during the test period, neither group had any significant adverse effects. These results show that taking supplements of LMT1-48 collaborates in reducing abdominal obesity without raising any safety concerns.

A recently discovered *L. plantarum* LMT1-48 has demonstrated weight-loss properties in preclinical research [27]. By downregulating lipogenic genes such as fatty acid synthase, fatty acid binding protein 4, peroxisome proliferator-activated receptor gamma (PPARγ), and CCAAT/enhancer binding protein, the intake of an LMT1-48 extract to 3T3-L1 adipocytes prevented their differentiation and lipid accumulation [28]. LMT1-48 decreased body weight and fat mass in a mouse study with fat mice given a high-fat diet [28]. Furthermore, a previous RCT in obese participants found weight loss and BMI reduction in those who took LMT1-48 [29]. However, our study further revealed a decrease in fat mass and body fat percentage, while lean body mass was maintained by using DEXA. The decrease in fat mass or percentage by LMT1-48 was monitored through all body parts including the trunk, android, gynoid, legs, and arms.

Regarding the results of the fecal samples, numerous studies have shown that administration of probiotics can improve obesity via modulating the gut microbiome. Our findings revealed a significant enrichment of certain microbial taxa, including *Hafnia*, *Lactobacillus*, *Lachnospira*, and *Phocaeicola*_A_858004, in the experimental group administered with *L. plantarum* LMT1-48 compared with the placebo group. Among these taxa, genera such as *Hafnia*, *Lachnospira*, and *Phocaeicola*_A_858004 have been previously linked to obesity [30,31,32]. Specifically, a previous study demonstrated that the commensal *Hafnia alvei* reduced body weight gain and fat mass in obesity models, as well as food intake in hyperphagic obese mice, through the production of ClpB protein—a bacterial heat-shock protein that antagonizes α-melanocyte-stimulating hormone (α-MSH), affecting the release of satiety-related hormones such as GLP-1 or Peptide YY in the intestine [31,33]. Furthermore, a negative correlation between the abundance of Hafnia with BMI has been reported [31]. Additionally, *Phocaeicola vulgatus*, which was recently reclassified from *Bacteroides vulgatus* [34], has been shown to alleviate high-fat diet-induced obesity by inhibiting intestinal serotonin synthesis, leading to reduced lipid absorption [35]. This taxon also contributes to impeding obesity by catabolizing branched-chain amino acids in brown fat [30]. In the case of *Lachnospira*, its abundance is significantly reduced in overweight/obese individuals compared with non-overweight/obese, consistent with our observations [36]. The increased abundance of *Lactobacillus* is likely attributed to the supplementation of a probiotic *L. plantarum* LMT1-48. Given that the administration of this novel probiotic influences obesity-associated microbial taxa, *L. plantarum* LMT1-48 holds potential as a novel probiotic to mitigate body weight gain and/or diet-induced obesity.

Neither participant group experienced any clinically significant side effects during the period of the 12-week investigation. The lactic acid-producing bacterium LMT1-48 was considered safe, as evidenced by the compliance level of >95% in both groups.

Although it has been observed that administering such probiotics improves the metabolic state, the results vary depending on the strain, combination, dose, and solvent utilized [29]. For example the intake of *L. reuteri* NCIMB 30242 (5 × 10^10^ CFU) in a yogurt formulation by healthy individuals [37] and *L. reuteri* JBD301 (1 × 10^10^ CFU) have been shown to reduce body weight. The degree of the effect is similar to orlistat, an authorized anti-obesity medication [38].

According to previous studies, various mechanisms were proposed for probiotics to help with obesity. Anti-obesity of probiotics is related to improvements in lipid metabolism, insulin sensitivity, anti-inflammation, or control of intestinal hormones like leptin or GLP-1, which is frequently accompanied by modulating the intestinal microbiota [39,40,41]. However, each probiotic strain employs a different mechanism to exert its anti-obesity effect, as each strain produces different metabolites [39]. Currently, the anti-obesity action of *L. plantarum* LMT1-48 is linked to its effect on lipid metabolism and gut microbiota, according to this and previous studies [28], but further studies may be necessary on metabolome change by LMT1-48 supplementation or on the effect on other biomarkers to understand the mechanism of this strain more precisely.

Our study has several limitations. Firstly, in clinical trials, probiotics cannot be handled like conventional medications, since they are live organisms. Probiotic strains can provide different outcomes based on factors including methods of storage and intake behaviors, unlike drugs, for which consistent results can be predicted through regular storage and consumption practices. Therefore, probiotics’ benefits can vary from person to person and can last only as long as they are taken [42]. Finally, although it is generally suggested that probiotics lose their effectiveness once users stop taking them [42], further research is needed to determine how long after discontinuation they lose their effectiveness (i.e., half-life).

## 5. Conclusions

In conclusion, our RCT has shown that LMT1-48 has a therapeutic effect on collaborating in reducing obesity in humans. Meanwhile, LMT1-48 is capable of modulating specific gut microbiota associated with obesity. This provides support to the notion that probiotic-based therapies aimed at modifying the gut microbiota hold considerable therapeutic promise for reducing obesity.

## Figures and Tables

**Figure 1 nutrients-17-01191-f001:**
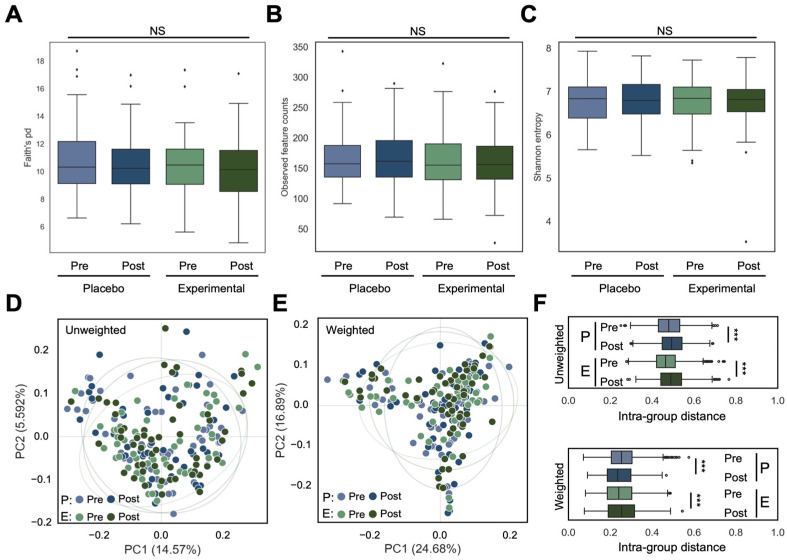
Compositional and structural changes in the gut microbial community pre- and post-treatment in both experimental and placebo groups. Alpha diversity was assessed using (**A**) Faith’s pd, (**B**) observed feature counts, and (**C**) Shannon entropy. Data shown and error bars are mean ± SD (Kruskal–Wallis test; NS, nonsignificant). Beta diversity was visualized with PCoA based on unweighted UniFrac distance (**D**,**E**) and weighted UniFrac distance. Statistical differences were evaluated with PERMANOVA. (**F**) Intra-group distances using unweighted (top) and weighted (bottom) UniFrac distances. Data shown and error bars are mean ± SD (non-parametric *t*-test, *** *p* < 0.001). “P” and “E” on the graph represent the placebo and experimental groups, respectively.

**Figure 2 nutrients-17-01191-f002:**
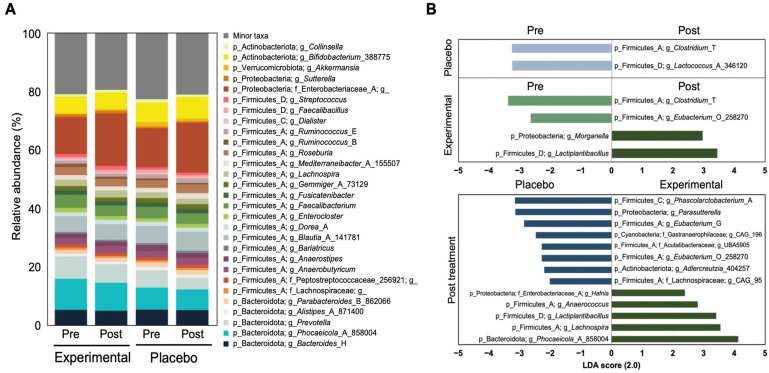
Alteration of gut microbial composition before and after probiotic treatment. (**A**) Relative abundance plot of bacterial taxa at the genus level according to probiotic intervention. (**B**) Bar chart denoting differentially abundant taxa according to probiotic treatment using LEfSe (LDA > 2). The LDA scores indicate the effect sizes of the taxonomy with the difference in relative abundances between pre- and post-treatment.

**Table 1 nutrients-17-01191-t001:** Baseline characteristics of the study population.

	LMT1-48 (*n* = 53)	Placebo (*n* = 53)	*p*-Value
Gender (Male)	21 (39.6)	18 (34.0)	0.546
Age	44.6 ± 12.9	40.7 ± 10.2	0.084
Height (cm)	166.3 ± 8.3	166.7 ± 9.3	0.780
Weight (kg)	75.5 ± 8.2	75.3 ± 9.6	0.762
BMI (kg/m^2^)	27.3 ± 1.3	27.0 ± 1.3	0.303
SBP (mmHg)	126.0 ± 13.4	117.9 ± 12.2	0.002
DBP (mmHg)	75.3 ± 9.9	71.8 ± 11.6	0.043
Glucose (mg/dL)	86.7 ± 8.3	85.8 ± 9.2	0.205
Total cholesterol (mg/dL)	208.8 ± 34.6	198.4 ± 30.5	0.104
Triglyceride (mg/dL)	136.9 ± 136.8	111.1 ± 57.4	0.825
HDL cholesterol (mg/dL)	55.6 ± 15.0	52.1 ± 9.3	0.546
LDL cholesterol (mg/dL)	126.3 ± 29.4	125.0 ± 25.2	0.796
AST (IU/L)	27.4 ± 12.4	24.3 ± 7.2	0.181
ALT (IU/L)	27.8 ± 17.9	26.9 ± 20.1	0.612
Creatinine (mg/dL)	0.75 ± 0.18	0.74 ± 0.13	0.918
Arm circumference (cm)	30.9 ± 2.3	30.3 ± 1.9	0.140
Waist circumference (cm)	57.4 ± 4.5	57.4 ± 3.5	0.960
Hip circumference (cm)	103.9 ± 3.8	104.4 ± 4.4	0.539
Waist/hip ratio	0.90 ± 0.06	0.91 ± 0.06	0.497
Smoking status			0.340
Never smoker	38 (71.7)	43 (81.1)	
Ex-smoker	6 (11.3)	2 (3.8)	
Current smoker	9 (17.0)	8 (15.1)	
Alcohol intake	31 (58.5)	32 (60.4)	0.843
Physical activity (MET value, h/week)	64.31 ± 67.84	53.10 ± 40.89	0.528
Body fat mass (kg)	30.0± 4.4	29.0 ± 4.4	0.286
Body fat percentage (%)	40.9 ± 6.8	39.7 ± 5.8	0.211
Lean mass (kg)	41.9 ± 8.5	42.5 ± 8.2	0.464
Visceral fat area (cm^2^)	129.7 ± 43.8	112.2 ± 34.5	0.043
Subcutaneous fat area (cm^2^)	236.8 ± 57.0	243.8 ± 57.2	0.533
Total abdominal fat area (cm^2^)	366.5 ± 66.0	356.0 ± 70.6	0.433

BMI, body mass index; SBP, systolic blood pressure; DBP, diastolic blood pressure; HDL, high-density lipoprotein; LDL, low-density lipoprotein; AST, aspartate aminotransferase; ALT, alanine aminotransferase; MET, metabolic equivalent of task.

**Table 2 nutrients-17-01191-t002:** Impact of LMT1-48 on anthropometric measurements.

	Group	Baseline	Week 6	*p*-Value ^†^	*p*-Value ^‡^	Week 12	*p*-Value ^†^	*p*-Value ^‡^
Weight (kg)	LMT1-48	75.5 ± 8.2	74.6 ± 8.4	<0.001	0.249	74.1 ± 8.2	<0.001	0.453
	Placebo	75.3 ± 9.6	74.8 ± 9.8	0.021		74.2 ± 9.9	<0.001	
BMI (kg/m^2^)	LMT1-48	27.3 ± 1.3	26.9 ± 1.4	<0.001	0.186	26.7 ± 1.3	<0.001	0.421
	Placebo	27.0 ± 1.3	26.8 ± 1.5	0.016		26.6 ± 1.6	<0.001	
Arm circumference	LMT1-48	30.9 ± 2.3	30.8 ± 2.3	<0.001	0.497	30.7 ± 2.3	<0.001	0.342
	Placebo	30.3 ± 1.9	30.2 ± 1.9	0.135		30.1 ± 1.8	0.023	
Waist circumference	LMT1-48	94.5 ± 4.6	93.8 ± 4.6	<0.001	0.318	93.5 ± 4.5	<0.001	0.452
	Placebo	94.2 ± 5.9	93.8 ± 5.9	0.015		93.3 ± 6.2	<0.001	
Hip circumference	LMT1-48	103.9 ± 3.8	103.5 ± 3.6	0.002	0.173	103.3 ± 3.5	<0.001	0.548
	Placebo	104.4 ± 4.4	104.2 ± 4.3	0.208		104 ± 4.2	0.034	
Calorie intake (kcal)	LMT1-48	1637.8 ± 407.4	1355.4 ± 380.4	<0.001	0.294	1304.5 ± 426.8	<0.001	0.139
	Placebo	1639.4 ± 405.0	1309.1 ± 389.7	<0.001		1223.5 ± 386.1	<0.001	
Physical activity (MET, hr/week)	LMT1-48	64.31 ± 67.84	67.09 ± 51.55	0.789	0.705	65.82 ± 39.19	0.880	0.654
	Placebo	53.1 ± 40.89	58.43 ± 48.01	0.391		62.79 ± 47.46	0.173	

^†^ Compared within groups; ^‡^ Compared between groups.

**Table 3 nutrients-17-01191-t003:** Impact of LMT1-48 on radiologic measurements and blood biochemistry profiles.

	Group	Baseline	Week 12	*p*-Value ^†^	*p*-Value ^‡^
Body fat mass (kg)	LMT1-48	30.0 ± 4.4	28.3 ± 4.1	<0.001	0.009
	Placebo	29.0 ± 4.4	28.3 ± 4.3	0.020	
Body fat percentage (%)	LMT1-48	40.9 ± 6.8	39.4 ± 6.3	<0.001	0.004
	Placebo	39.7 ± 5.8	39.3 ± 5.8	0.229	
Lean mass (kg)	LMT1-48	41.9 ± 8.5	42.1 ± 8.1	0.388	0.026
	Placebo	42.5 ± 8.2	42.2 ± 8.4	0.061	
Visceral fat area (cm^2^)	LMT1-48	129.7 ± 43.8	120.9 ± 43.7	<0.001	0.368
	Placebo	112.2 ± 34.5	107.0 ± 33.4	0.017	
Subcutaneous fat area (cm^2^)	LMT1-48	236.8 ± 57.0	222.2 ± 61.8	<0.001	0.185
	Placebo	243.8 ± 57.2	236.2 ± 64.7	0.055	
Total abdominal fat area (cm^2^)	LMT1-48	366.5 ± 66.0	343.1 ± 73.3	<0.001	0.116
	Placebo	356.0 ± 70.6	343.2 ± 81.2	0.012	
Glucose (mg/dL)	LMT1-48	86.7 ± 8.3	86.3 ± 8.5	0.354	0.116
	Placebo	85.8 ± 9.2	86.1 ± 9.7	0.146	
Total cholesterol (mg/dL)	LMT1-48	208.8 ± 34.6	205.8 ± 31.8	0.417	0.249
	Placebo	198.4 ± 30.5	191.9 ± 30.9	0.028	
Triglyceride (mg/dL)	LMT1-48	136.9 ± 136.8	105.6 ± 68.2	0.048	0.995
	Placebo	111.1 ± 57.4	94.3 ± 51.1	0.006	
HDL cholesterol (mg/dL)	LMT1-48	55.6 ± 15.0	55.3 ± 14.1	0.764	0.660
	Placebo	52.1 ± 9.3	51.3 ± 8.9	0.280	
LDL cholesterol (mg/dL)	LMT1-48	126.3 ± 29.4	128.9 ± 28.2	0.429	0.114
	Placebo	125.0 ± 25.2	122.4 ± 25.6	0.307	
Free fatty acid (μmol/L ^c^)	LMT1-48	587.6 ± 315.0	596.2 ± 212.8	0.830	0.471
	Placebo	545.1 ± 219.6	624.3 ± 298.6	0.051	
Adiponectin (μg/mL)	LMT1-48	4.08 ± 0.84	4.85 ± 0.56	<0.001	0.540
	Placebo	4.11 ± 0.81	4.87 ± 0.52	<0.001	

^†^ Compared within groups; ^‡^ Compared between groups; μmol/L ^c^, micromoles per litre concentration (i.e., number of micromoles in 1 litre of solution).

**Table 4 nutrients-17-01191-t004:** Body fat mass and body fat percentage by body part.

**Body Fat Mass**	**Group**	**Baseline**	**Week 12**	***p*-Value ^†^**	***p*-Value ^‡^**
Head (kg)	LMT1-48	1.3 ± 0.1	1.3 ± 0.1	0.501	0.259
	Placebo	1.3 ± 0.1	1.3 ± 0.2	0.037	
Trunk (kg)	LMT1-48	15.4 ± 2.2	14.4 ± 2.2	<0.001	0.013
	Placebo	14.8 ± 2.6	14.4 ± 2.5	0.001	
Android (kg)	LMT1-48	2.6 ± 0.5	2.4 ± 0.4	<0.001	0.041
	Placebo	2.5 ± 0.6	2.4 ± 0.6	0.015	
Gynoid (kg)	LMT1-48	4.8 ± 1.0	4.6 ± 0.9	<0.001	0.011
	Placebo	4.7 ± 1.0	4.6 ± 0.9	0.030	
Rt. arm (kg)	LMT1-48	1.8 ± 0.4	1.8 ± 0.3	0.012	0.098
	Placebo	1.7 ± 0.3	1.7 ± 0.3	0.806	
Lt. arm (kg)	LMT1-48	1.8 ± 0.4	1.8 ± 0.4	0.032	0.152
	Placebo	1.7 ± 0.3	1.7 ± 0.3	0.835	
Rt. leg (kg)	LMT1-48	4.8 ± 1.2	4.5 ± 1.0	<0.001	0.042
	Placebo	4.7 ± 1.1	4.5 ± 1.0	0.007	
Lt. leg (kg)	LMT1-48	4.8 ± 1.1	4.5 ± 1.0	<0.001	0.005
	Placebo	4.7 ± 1.1	4.6 ± 1.0	0.122	
**Body Fat Percentage**	**Group**	**Baseline**	**Week 12**	***p*-Value ^†^**	***p*-Value ^‡^**
Head (%)	LMT1-48	26.5 ± 0.4	26.5 ± 0.5	0.547	0.370
	Placebo	26.3 ± 0.4	26.4 ± 0.5	0.177	
Trunk (%)	LMT1-48	42.3 ± 6.1	40.5 ± 5.9	<0.001	0.011
	Placebo	40.9 ± 5.4	40.5 ± 5.4	0.289	
Android (%)	LMT1-48	45.4 ± 5.7	43.4 ± 5.3	<0.001	0.028
	Placebo	44.1 ± 5.3	43.5 ± 5.2	0.073	
Gynoid (%)	LMT1-48	41.5 ± 7.9	39.8 ± 7.1	<0.001	0.005
	Placebo	40.5 ± 7.1	40.1 ± 7.0	0.159	
Rt. arm (%)	LMT1-48	46.0 ± 10.9	44.8 ± 10.2	0.001	0.026
	Placebo	45.6 ± 9.2	45.5 ± 9.1	0.770	
Lt. arm (%)	LMT1-48	44.9 ± 10.5	43.7 ± 10.0	0.002	0.003
	Placebo	43.4 ± 8.4	43.8 ± 8.9	0.301	
Rt. leg (%)	LMT1-48	40.4 ± 8.9	38.8 ± 7.8	<0.001	0.043
	Placebo	39.1 ± 7.8	38.5 ± 7.8	0.067	
Lt. leg (%)	LMT1-48	39.5 ± 8.5	37.9 ± 8.0	<0.001	0.003
	Placebo	38.7 ± 7.8	38.2 ± 7.6	0.081	

^†^ Compared within groups; ^‡^ Compared between groups.

## Data Availability

For data supporting the reported results, contact the corresponding author. The data are not publicly available due to privacy.

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
