# Peer review of "A 12-Week, Randomized, Double-Blind, Placebo-Controlled Study to Evaluate the Efficacy and Safety of Lactobacillus plantarum LMT1-48 on Body Fat Loss"

_nutrients, 2025, doi:10.3390/nu17071191_

Round 1

Reviewer 1 Report

Comments and Suggestions for Authors

Reviewer 2 Report

Comments and Suggestions for Authors

General comments:
Overall, the manuscript reflects a trial conducted to test the efficacy of a probiotic (L. plantarum LMT1-48) as an adjunctive medication to aid in weight loss and body adiposity. Overall, the number of subjects employed in the study (106) is significant, as is the number of parameters studied. The exclusion factors included are very comprehensive and overall, with some minor errors, the manuscript is reasonably well written and presented. Perhaps the biggest flaw I see in it is that since the participants were instructed to reduce their caloric intake and increase energy expenditure, the observed effects cannot be fully attributed to the probiotic, as the behavior of the participants has been modified. Therefore, I believe that the attribution of the observed effects to the probiotic should be relaxed.

Specific comments:

Title: names of bacterial species must be written in italics
Abstract sections. Does not make sense to use abbreviations such as “DEXA” or “CT” that only appears once in the abstract.
Lines 21-24: This paragraph is absolutely expendable.
Lines 44-50: This paragraph is not in the same format as the other text.
Line 73: The names of bacterial species must be written in italics throughout the manuscript.
Page 3 is blank, please delete this page
Lines 104-105 “®” must be in superscript
Line 112: once a day for how may days?
Lines 145-146: “At 145 visits 2 and 3, all participants were educated to reduce food intake by 500 kcal per day 146 and to increase daily physical activity by 300 kcal during study”. this is important. I know that there are many previous works that perform a similar experimental design, but then the results obtained cannot be attributed to the effect of the probiotic, but rather that the probiotic is simply a supplement that would help. Many expressions in the article that attribute the effects obtained to the L. plantarum used should be rephrased.
Lines 149-174: It should be mentioned the equipment used in all anthropometric measurements and serum determinations 
Line 229-230: “no significant difference in consumption” what consumption? do not refer to anthropometric data?
Line 233: Please define “SBP” and “DBP”.
Table 2 is in a different font size than other tables, and must be placed in an only sheet.
Line 302: In all manuscript significant levels were cited as “p” and not “P”. Please, be consistent.
Lines 307 and afterwards: Both genus and species bacterial names must be written in italics.
Line 338: “demonstrated weight loss properties” This is an example of phrase that should be rewritten. If the volunteers were instructed to reduce caloric intake and increase physical activity, you cannot attribute the weight loss observed in the subjects to the action of Lactobacillus plantarum. At most you can say that this lactobacillus “collaborated” in this weight loss.
Line 395-397: “External influences, such as holidays and private vacations, may have caused participants to consume more calories and engage in less physical exercise, making it difficult to maintain consistent calorie intake due to uncontrollable circumstances”. Are you aware that this event occurred during the trial conducted? if not, it cannot be said to be a limitation of the work itself and it is better to eliminate it.
Line 404: in collaborating alleviating
Line 410: “and disorders” no any disorder was alleviated in this assay. Please reduce to obesity and not associated disorders.
Supplementary information (tables A1-A3) and Figure A1 must be placed separately from the main text in another file.
Please be more careful with references style. In example, reference 5 is out of format. Reference 6 is incomplete…etc.
